# The Participatory Budgeting and Its contribution to Local Management and Governance: Review of Experience of Rural Communities from the Ecuadorian Amazon Rainforest

**Irene Buele** [1,2,*] , **Pablo Vidueira** [1], **José Luis Yagüe** [1] **and Fabián Cuesta** [2]

1   Escuela Técnica Superior de Ingeniería Agronomica, Alimentaria y de Biosistemas, Universidad Politécnica de Madrid, 28040 Madrid, Spain; pablo.vidueira@upm.es (P.V.); joseluis.yague@upm.es (J.L.Y.)
2   Sede Cuenca, Universidad Politécnica Salesiana, 010102 Cuenca, Ecuador; fcuesta@ups.edu.ec
*   Correspondence: ig.buele@alumnos.upm.es

**Abstract:** In Ecuador, the participatory political design of the political party forming the government from 2007 to 2017, along with the constitution of 2018, created opportunities for citizen participation. Participatory budgeting (PB) is the most commonly used citizen participation mechanism. The direct participation of citizens is reflected in improving the governance by democratizing decision processes. The contribution of PB to the local management and governance of seven rural communities of the Ecuadorian Amazon was analyzed using a case study. Based on (1) the level of compliance with municipal planning through management indicators and, the amounts allocated to PB, (2) along with the level of citizen satisfaction, complementary perspectives (acquired through a survey) on the implementation of PB are provided. These sources of evidence allowed us to critically assess the effects of PB in the improvement of local management and governance. We found low levels of municipal planning compliance, i.e., a 20% (2017) and 43% (2018), high levels of citizen dissatisfaction (around 91%) and also a "disagreement" with the PB implementation process. Finally, it is observed that the implementation of the participatory budget in rural communities presents deficiencies that limit the obtaining of representative benefits and that imply an improvement in the governance and quality of life of the citizenry. This is mainly caused by the low interest of citizens to participate in the phases of execution and monitoring of projects, due to a low culture and participatory education.

**Keywords:** participatory budgeting; multi-level governance; local government; rural communities

## 1. Introduction

In Ecuador, the 2018 Constitution institutionalized citizen participation and social control under the label of "participation rights" [1]. These include the rights of citizens to freely engage in intercultural, inclusive, diverse, and participatory communication. For Gutiérrez, this new Constitution signals a change for the country amid discontent, instability, and the rise of the idea of citizen power as a counterweight to political parties—ideas promoted by Rafael Correa through his political party, Alianza País [2]. In this context, the new citizen participation system was launched. In 2010, the Organic Law of Citizen Participation (abbreviated in Spanish as LOPC) was enacted, establishing participation rights and indicating the obligation to present annual budgets for all levels of government within the framework of an open call for citizen participation, which is an obligation to implement participatory budgets (PB). In Ecuador, the law requires that citizens be included in the budget formulation; however, there is no law that states and details the process of participatory budgeting implementation, only general guidelines and instructions have been established. Each Municipal

Decentralized Autonomous Government (DAG) is free to design its own process through a municipal ordinance. Therefore, the application of PB as a structured and formal process is applied only in some municipalities in Ecuador.

Citizen participation constitutes an element inherent to democracy and to advanced democratic governance environments, and implies the intervention of citizens in government [3]. This citizen participation is the result of the level of development, multiple social interactions, changes in systems politicians, the production of networks at global, national and local scales, the emergence of international organizations, globalization, the new instruments of public policy control, the formation of societies that increasingly become more critical and demanding of their rulers, among others [4].

Governance, then, is currently linked to the concept of citizen participation, whose application seeks increased efficiency and efficacy which translates into effectiveness and transparency in public management [5]. The incorporation of horizontal–territorial networks changes the form of policy production aimed to consolidate local governance so that participation is legitimized [6]. Democratically making decisions and developing a sense of responsibility seem to be the best path to governance [7]. Therefore, the engagement of governments with citizens in democratic decision making and shared responsibilities can meaningfully strengthen governance, even though consensus may be difficult to reach.

Gómez-Hernández defined governance as the democratization of decision-making processes on territorial development through the direct participation of citizens in consultation with political, public, and private actors for the configuration and ordering of political, social, economic, and cultural localities [8]. This concept was born from a process of democratic evolution. In the 1980s and 1990s, this process mainly questioned the effectiveness of governments in deciding, managing, and implementing policies without interference from society and politics. In 2000, governance acquired a broader meaning. In addition to stability and efficiency, the favoring of government by the citizens reinforces the democratic spirit to the processes [9]. Navarro, Veneziano, and Rojas emphasized the concept of governance is only achieved if the state is able to articulate the participation of the various sociopolitical actors in the design and implementation of public policies and in institutional design based on the awareness of the rights of citizens to demand transparency [6,7,10]. The United Nations has also highlighted the importance of combining peace, sustainable development, and effective governance through accessible and accountable institutions [11].

The governance analysis involves both the institutional functioning and the relationship of citizens with the institutions and other practices of citizen participation [12]. In municipalities throughout Latin America, citizen participation in planning and budgeting has become an indicator of good governance [13], or at least a tool [14] to generate governance [7]. With the application of PB, the government seeks to strengthen social capital and obtain legitimacy through procedures necessary to promote governance [6]. The implementation and subsequent evaluation of PB are key elements of good governance [15]. For Arzaluz, one of the conditions required for participatory planning in order to develop and produce results is the agreement of different actors on specific objectives, on the procedures, and on the rules that allow a fluid relationship between those involved in the process [16]. As an additional condition, Alguacil includes the importance of having trained and motivated public employees who have a public vocation to help create a community of citizens with sufficient social cohesion, the establishment of institutional commitments in matters of citizenship, and a vocation of continuity through the establishment of institutional mechanisms [17].

PB is considered as a tool that can enhance and strengthen democratic participation and community development, based on the involvement of citizens in the identification of priority needs [18]. Through it, investments can be made by governments that fit with key societal priorities and boost development at the local level [19]. Martínez and Arena [20] and Sgueo [21] stated that PB represents a substantial commitment to expanding and improving the rule of law and the democratic system.

Baiocchi and Ganuza [22] and Sgueo [21] agree that PB starts from the idea that ordinary citizens should be part of the codecision process of the public budget which has a significant impact on them.

Martínez and Arena stated that PB represents a substantial commitment to expanding and improving the rule of law and the democratic system [20]. PB has the potential to rebuild relationships between the government and communities, since joint work is promoted [23]. Although this objective is achieved in most cases, some authors mention that getting to improve this relationship is not always possible [24].

One of the aims of PB is to improve the relationship between local governments and the population. A participatory budget requires communication between community organizations and their leaders to work collectively [25] to improve local management and achieve a better redistribution of resources, citizen participation, and political transparency [26]. PB is expected to reduce conflicts and engender a wider acceptance of budgetary decisions [21].

In an ideal situation, PB influences changes in the living conditions of the population and in the infrastructure of the towns based on territorial balance and the criteria of distributive justice [6]. However, according to Pagani, a risk always exists that the technical evaluation of municipal experts will weaken popular participation—that calls will be insufficient, and the participation will not be sufficiently representative to produce large changes in the population [5]. These limitations may depend on political will, a factor that has been considered a limitation for the implementation and development of PB in Latin America [27]. Kuruppu et al. considered the use of PB to reinforce the careers of local politicians and allocate resources and projects only to meet the requirements of their supporters as drawbacks [28].

However, determining the level of success of PB is complex, the evaluation represents a challenge in development; it is a continuous commitment to learning, reflection, responsibility, and correct decision-making [29]. Allegretti et al. stated that measuring the level of satisfaction and expectations of citizens according to the results obtained is important; if the immediate results of the participatory process do not align with the expectations of citizens, frustration is expressed, resulting in a loss of motivation, which jeopardizes future participation [30]. However, when frustration is absent, satisfied and committed citizens commonly disseminate the process, thereby increasing its popularity and legitimacy. Aedo stated that for citizens to be satisfied, they must engage in collective empowerment based on (1) leadership, ideology, and the concept of participation; (2) the level of transversality and the interconnection of administrative bodies; (3) the methodological model and participation in the design [31]. According to Simonsen and Robbins, the inclusion of citizens in the political processes and local government decisions is challenging, as citizens have their own opinions and may distrust governments [32].

PB has been extensively researched. According to Buele and Vidueira, 37% of publications, related to PB in the Redalyc, Web of Science and Scopus databases from 2000 to 2016, focus on studies at the local level, referring to municipal experiences, governance, and citizen participation, among other elements; 25% are conducted at the regional and national levels, referring to models of budgets, effects, citizen preferences, etc.; 38% of the investigations focus instead to a broader field, such as worldwide studies and the general or theoretical applications of PB [33]. PB has been more widely applied in urban contexts, as a mechanism for modernizing local management and governance by facilitating people's participation in budget decisions [29]. In rural communities, people's participation tends to focus on the selection and operation as basic services [34] such as drinking water and sewerage systems, but not that much on modernizing local management and governance as in the urban areas. However, research and publications regarding the application of PB in rural environments in Latin America, where the population has special characteristics and particularities, are scarce.

This case study is carried out in Ecuador. It adopts the categorization made by the Economic Commission for Latin America (ECLA) that distinguishes between rural and urban cities. This institution considers as urban locations all those settlements that are administrative headquarters of the municipalities and defines as rural areas the remaining ones. Thus, three perspectives are recognized: (1) Chile, Costa Rica, Cuba, Panama, and Uruguay focus on population density and the availability of public services; (2) Argentina, Bolivia, Guatemala, Mexico, and Venezuela established a minimum population of 2000 inhabitants as a standard for a town to be considered urban; (3) Brazil, Colombia,

Ecuador, El Salvador, Haiti, Honduras, Nicaragua, Paraguay, Peru, and the Dominican Republic define an urban location as a city that is the administrative headquarters of the municipality and includes populations of less than 2000 inhabitants (or 250 households), without any other requirement for the availability of basic services [35]. In Ecuador, the administrative seat of the municipality is considered an urban area without considering the number of inhabitants or any other requirement, while the remaining cities and communities that are not the administrative seat of the municipality are classified as rural areas. Ecuador has an active rural population of 2,192,593 inhabitants, which represents 34% of the active economic population; 57% of this population works in the Sierra, 34% in the Coast, and 9% in the Amazon [36].

In this study, we critically analyzed the implementation of PB in seven rural Amazonian communities. We integrated a review of the level of compliance of municipal planning (by applying management indicators), the level of citizen satisfaction and complementary perspectives on the implementation of PB, which are factors that allow to analyze the contribution of PB to the processes of management and governance in a rural municipality.

*Case Study*

Ecuador is a country located in South America. Geographically, it is divided into four natural regions: the Coast, the Sierra, the Amazon, and the Galapagos Archipelago. Ecuador is characterized by the isolation of many of its communities [37], poverty, and a general lack of resources [38]. Poverty is centered in rural areas, where 50.2% of the population live below the poverty line, and 25.6% are considered poor [39]. Ecuador has an active rural population of 2,192,593 inhabitants [36], the highest rates of poverty are found in the Amazon at 59.7%, along the Coast at 40.3%, and in the Sierra at 33.7%.

This study was conducted in the municipal Decentralized Autonomous Government (DAG) of the canton of Gualaquiza, in the province of Morona Santiago, one of the provinces of the Amazon region of Ecuador. A DAG is a legal entity under public law with political, administrative, and financial autonomy. Is integrated into the functions of citizen participation, legislation, and executive enforcement, and can be regional, provincial, cantonal (municipal), or parochial (communities) [40]. The DAG studied in this research is cantonal and responsible for planning the development of the canton and formulating territorial planning in conjunction with national, regional, provincial, and district planning to regulate the use of urban and rural land [41]. Among its integrated functions are citizen participation and social control, which are necessary to implement a system of citizen participation to exercise the rights and democratic management of municipal action [42].

The Organic Law of Citizen Participation of 2010 formalizes the practice of PB for municipal DAGs by indicating the obligation to present annual budgets for all levels of government, within the framework of an open call for citizen participation [43]. This law mentions the procedure for the preparation of PB, which can be summarized into three phases: (1) public deliberation of the formulation of budgets; (2) discussion and approval of the topics with citizens, social organizations, and delegates for the basic units of participation, communities, communes, enclosures, neighborhoods, urban, and rural parishes, in decentralized autonomous governments; (3) monitoring budget execution. This is supported by the invention and creation of new models of local management with citizen participation. The Constitution of 2008 and the Organic Law of Citizen Participation of 2010 encouraged municipal DAGs to involve citizens in the realization of their budget—a mechanism known as PB. However, during the first years, few municipalities knew about this mechanism, and those that applied it were called alternative municipalities [44]. As explained above, the law does not indicate a detailed and formal process to apply PB, which is the reason why each municipality has interpreted and applied what the law indicates according to its criteria.

The municipal DAG of Gualaquiza has an area of 2151.29 km$^2$, an average altitude 850 m above sea level, and a temperature that ranges between 22 and 27 °C [37]. This municipal DAG has 10 districts which are referred to in this article as communities: (1) Gualaquiza, (2) Mercedes Molina, (3) Bomboiza, (4) Nueva Tarqui, (5) El Rosario, (6) San Miguel de Cuyes, (7) Chigüinda, (8) Amazonas, (9) Bermejos,

and (10) El Ideal. Gualaquiza has 15,288 inhabitants, of which 7512 are female and 7776 are male; 58.6% (8,952 inhabitants) of its population resides in rural areas. This DAG is characterized as a young population, since 56.4% are under the age 20. A large proportion of the population, 25.7%, is dedicated to agriculture and livestock, with few engaged in manufacturing activities (4%). The 39.6% of the population is supplied with water by rivers or watersheds; 32.3% use wood as fuel for cooking; 43.8% of the population has sewerage; and 69.7% of the rural population has obtained a primary education level [39].

This municipal DAG did not apply PB until 2017 and 2018. In 2019, after a change of government, PB was not implemented again. Although the application of PB is required by law, each municipality performs it according to its criteria. The law does not indicate a formal and detailed application procedure, so some municipalities apply it superficially. In 2017, the municipal DAG of Gualaquiza approved a total budget of USD 11,538,432, and of this funding, 2% (a total of USD 240,986.08) was assigned to PB. For 2018, the DAG had a total budget of USD 12,139,197.38, and USD 480,546.08 (4%) was allocated to PB.

The basis for the implementation of PB is compliance with the provisions of the law. Within these confines, complying with the principles of citizen participation is essential as a minimum guarantee of the will of the municipal DAG to achieve success. Ecuadorian regulations indicate the principles of citizen participation that must be followed: (1) "Equality": Rabossi defined equality as 'Human beings should be considered and treated in the same way, that is, in a uniform and identical manner, unless there is a sufficient reason not to do so' [45] (p. 176), which is the reason why the PB implementation must respect this principle in all its phases. (2) "Gender Parity": this 'refers to relative equality in terms of numbers and proportions of women and men, girls and boys, and is often calculated as the proportion of values of women to men for a given indicator' [46]. It was observed that the municipal DAG of Gualaquiza assigned its budget independently to the gender of its inhabitants. Resources are allocated according to the number of men and women in each community, without giving preference to any gender, therefore, the correlation of the budget with each gender is positive. Therefore, the correlation of the budget with each gender is positive with values of 82.05% between the budget and men and 80.84% between the budget and women [36]. (3) "Respect for Differences": in this area, people with physical disabilities predominate over those with mental disabilities. In the municipal DAG of Gualaquiza 2.3% of the population has a physical disability [47]. (4) "Interculturality and Plurinationality": interculturality 'describes a set of multifaceted interaction processes through which relationships between different cultures are built, with the aim of allowing groups and individuals to forge links between cultures based on equity and mutual respect' [48] (p. 1). The Constitution of the Republic recognizes Ecuador as an intercultural and plurinational country. The study area is identified as 67.1% mestizo, 27% indigenous, and 3.4% white [40]. (5) "Responsibility and Co-Responsibility": these principles are met when implementing PB. The planning, organization, and execution of PB implies the fulfillment of its responsibilities as a municipal DAG. The citizenship demonstrates their responsibility and coresponsibility by attending the planned assemblies. (6) "Autonomy": the Ecuadorian Constitution grants decentralized autonomous governments political, administrative, and financial autonomy. This autonomy includes the right and effective capacity to be governed by rules and governmental bodies of and in their own territorial districts, under their responsibility, and without the intervention of another level of government, for the benefit of its inhabitants. According to the Organic Code of Territorial Planning Autonomy, article 5, political autonomy is the capacity of each decentralized autonomous government to promote processes and forms of development according to the history, culture, and characteristics of the territorial circumscription. Administrative autonomy involves fully exercising the faculty of organization and management of human talents and material resources to exercise citizens' competences and allow them to provide their contribution. Financial autonomy is expressed as the right of decentralized autonomous governments to directly receive predictable, timely, automatic, and unconditional resources [41]. (7) "Public Deliberation": the works defined in PB were discussed and approved through the participation of the inhabitants

of the communities in the citizen assemblies. This dialogue occurred in the last quarter of the year prior to the execution of PB; subsequently, the municipal DAG of Gualaquiza prepared the minutes of the executed assemblies. The assemblies are conducted with the following plan: welcoming remarks from the district and mayor representatives, explanation of the resource allocation methodology, presentation of the district's requirements, analysis and project prioritization, and closing statements. (8) "Popular Control, Information, and Transparency": these principles are key for citizens to trust the PB implementation process.

## 2. Materials and Methods

We critically analyzed the implementation of PB based on a review of two main components: (1) the level of compliance with planning through management indicators, and (2) the satisfaction and perception of citizenship through the application of surveys.

We evaluated the level of compliance with the projects planned through PB for 2017 and 2018 using indicators of efficiency and efficacy. The source of information corresponds to the institutional files of the municipal DAG of Gualaquiza. The Contraloría General del Estado, an institution that controls the use of public resources in Ecuador, notes that the State must be efficient and effective in its use of resources [49]. Management indicators include the units of organizational measurement used to evaluate performance before goals and objectives [50], and can also be used as guides to make strategic decisions that affect the future direction of an institution [51]. An efficiency indicator was used to measure the relationship between the resources used and the production of goods and services, expressed as a percentage comparing the resource–production relationship with an acceptable standard or norm. An efficacy indicator measured the relationship between the services or products that have been developed and the objectives that were planned [49]. Through a descriptive analysis, the factors that influenced the efficiency and efficacy of planning can be identified. In Ecuador, activities that include projects or development programs are known as "works". For the application of the indicators mentioned in Table 1, several sources of information and documents were requested from the municipal DAG of Gualaquiza, including: budget, budget execution report, files of the financial department, minutes of the assemblies and photographic file.

**Table 1.** Management indicators applied in this study to evaluate compliance with municipal planning.

| Indicator | Equation |
|---|---|
| Percentage of works carried out using PB. | (Executed works/Planned works) × 100 |
| Percentage of communities with budget allocation as a result of PB | (Communities with PB/All communities) × 100 |
| Percentage of municipal budget spent on works through PB | (Cost of work executed by PB/Total budget of the municipality executed) × 100 |
| Percentage of money spent on works executed through PB | (Cost of work executed by PB/Budget for each work by PB) × 100 |
| Amount of money spent on works executed through PB per inhabitant | (Cost of works executed through PB/Total inhabitants) |
| Percentage of total efficiency obtained in the execution of PB | Percentage of works carried out using PB × (Budget by PB/Cost of works) × 100 Less than 100% is less efficient than scheduled Equal to 100% meets the minimum expectations [52] |

Source: The authors.

The percentage indicator of works conducted through PB was calculated for the total of each item within a subprogram, and these results were averaged to obtain data for each subprogram and program, according to the budget and budget execution reports of the municipal DAG. The indicator of the percentage of money spent on works executed through PB was obtained by applying the formula

only for works that were executed, without considering the planned works, and these results were averaged to obtain information by subprogram and program.

To analyze the satisfaction and perception of citizens in the participating communities, an open survey was administered. This survey included 2 parts. The first part related to citizen's satisfaction with the implementation of PB and their perceptions regarding the inclusiveness of the process. The second part includes questions regarding the general phases of planning, execution and monitoring, rated according to the Likert scale. The sampling formula included a population (*N*) of 8952 inhabitants from rural areas [36], a confidence level of 90% (Z = 1645), and an error of 10%, which resulted in 68 surveys. These surveys were applied according to the weighting of the number of inhabitants per community, by random sampling and the data were extracted through personal surveys and by telephone. These surveys were carried out by telephone in communities too distant from the urban center, some of them requiring 3 to 5 hours by vehicle and then up to 3 hours of riding a horse to arrive at the populated center, which is located the only place with the Internet.

The survey was carried out using the Likert scale consisting of five response options: "Totally disagree" (1), "Disagree" (2), "Indifferent" (3), "Agree" (4), "Totally agree" (5). The information was tabulated and analyzed using the SPSS statistical analysis software. The reliability of the information was determined using the Cronbach's alpha coefficient that measures the consistency of a scale [53]. Cronbach's alpha establishes that the value obtained after its application to be acceptable must be between 0.8 and 1, where values close to 1 indicate that the items are reliable [54]. The survey had a reliability of 0.90.

Semistructured interviews with the local authorities were also used to help explain the numerical data. The interviews were conducted with the mayor, financial director, community leaders (elected by popular election) and technicians of the municipality responsible for planning, execution and monitoring PB. These semistructured interviews were carried out in parallel with the surveys and they inquired about positives/negatives, limitations, and purpose and satisfaction of the PB implementation process. The results were also analyzed using the correlation coefficient that measures the relationship between two variables. The correlation can range from +1 to −1, with 0 being the absence of correlation. If the correlation value is from 0.81 to 0.99, it qualifies as "High intensity"; if it is from 0.60 to 0.8 it qualifies as "Medium–High intensity"; if it is from 0.01 to 0.20 it qualifies as "Low intensity" [55].

## 3. Results

The results indicate the contribution of PB to the processes of management and governance from the following aspects: (1) the level of compliance of municipal planning, and (2) citizen satisfaction and perceptions.

### 3.1. Compliance Level by Means of Management Indicators

Neither the national regulations nor the municipal DAG of Gualaquiza have established a minimum amount that should be allocated. The allocation of 2% and 4% of PB are considered within the regional limits, which ranges between 2% and 15% of the total budget, according to national and international experiences [56]. Garrido and Montencinos strengthened this criterion by mentioning that in Chile, the percentages allocated to participatory budgeting do not exceed, on average, 3% of the total municipal budget [57].

According to Table 2, in 2017, the largest number of works, through PB, focused on the potable water subprogram, while the works of the entire budget of the municipality were directed mainly to the subprogram of "Construction and Maintenance Works and Urban Roads". In 2018, the works determined by PB were mostly construction and maintenance works and works related to urban roads, and the works of the entire budget of the municipality were also directed mainly to this subprogram. These subprograms respond to the needs of the population and are classified by the Economic Commission for Latin America (the Spanish acronym is CEPAL) as basic needs [58]. In addition,

the Amazon sector is characterized by having a weak road infrastructure [59]. In 2018, the number of works planned using PB increased by 32% (22 works for 2017 and 29 works for 2018).

**Table 2.** Subprograms in which the PB of the municipal DAG of Gualaquiza is distributed.

| Item | Year 2017 | | | | Year 2018 | | | |
|---|---|---|---|---|---|---|---|---|
| | Initial Assignment (USD) | Assignment Percentage | No. Works | No. Communities | Initial Assignment (Dollars) | Assignment Percentage | No. Works | No. Communities |
| Municipal DAG of Gualaquiza PB | 240,986.08 | 100% | 22 | | 480,546.08 | 100% | 29 | |
| Subprogram Potable Water | 99,998.84 | 42% | 11 | 5 | 82,680.23 | 17% | 9 | 4 |
| Subprogram Pluvial and Sanitary Sewage | 68,482.00 | 28% | 6 | 4 | 179,406.86 | 37% | 8 | 7 |
| Subprogram Construction and Maintenance Works and Urban Roads | 52,103.24 | 22% | 4 | 2 | 218,458.99 | 45% | 12 | 4 |
| Subprogram Collection Solid Waste and Final Disposition | 20,402.00 | 8% | 1 | 1 | 0.00 | 0% | 0 | 0 |

Source: PB of the municipal DAG of Gualaquiza.

According to the minutes of the assemblies held, a consensus is reached with 7 of the 10 communities of the municipal DAG of Gualaquiza. Table 2 shows that only the "Subprogram Pluvial and Sanitary Sewage" subprogram, in 2018, presented works for all communities. The subprogram "Construction and Maintenance Works and Urban Roads", to which the highest amounts correspond, is not only focused on works for urban communities but also includes works for remote communities with broad needs, such as construction of classrooms, design of wastewater treatments, construction of cemeteries and sports fields, among others. However, at the end of each year, only the construction processes of classrooms and sports fields were carried out.

Table 3 shows that in 2017, only 20% of the planned works were completed using PB, and these executed works consumed 97% of the resources designated for their realization. Of the total budget spent by the municipality, just 0.9% corresponded to works carried out through PB, and in total only 69% of what was planned was spent. In 2018, the situation improved: 43% of the planned works were completed, and these works consumed 99% of the budget designated for their realization. Of the total budget spent by the municipality, just 2% corresponded to works carried out through PB, and in total only 68% of what was planned was spent. For 2017, USD 31.27 of PB should have been allocated per inhabitant and USD 62.29 in 2018. However, values of USD 9.13 and USD 17.47, respectively, were observed. These results demonstrate the inefficiency of the execution of the planning of PB; that is, the municipal DAG of Gualaquiza has a low spending capacity. Based on the interviews carried out, this situation occurred due to the numerous bureaucratic processes involved in the execution of municipal works and the exhaustive coordination that is required between regional organizations [60]. There are many factors that make the application of PB difficult in rural areas: administrators' lack of interest in organizing and motivating the community; problems with the National System of Public Procurement; a lack of technicians; contractors' disinterest regarding being awarded small projects, together with citizens' lack of interest in asking their representatives to monitor planned works; long distances and poor connectivity among communities.

**Table 3.** Application of management indicators.

| Items | 2017 | | | 2018 | | |
|---|---|---|---|---|---|---|
| | Percentage of Works Completed Using PB | Percentage of Money Spent on Works Executed through PB | Amount Spent on Works Executed through PB Per Inhabitant | Percentage of Works Completed Using PB | Percentage of Money Spent on Works Executed through PB | Amount Spent on Works Executed through PB Per Inhabitant |
| Municipal DAG of Gualaquiza PB | 20% | 97% | 9.13 | 43% | 99% | 17.47 |
| Environmental Management | 0% | 0% | | 0% | 0% | |
| Subprogram Collection Solid Waste and Final Disposition | 0% | 0% | | 0% | 0% | |
| Infrastructure Works | 0% | 0% | | 0% | 0% | |
| Public Services Management | 27% | 94% | | 61% | 99% | |
| Subprogram Potable Water | 54% | 94% | | 59% | 100% | |
| Infrastructure Works | 33% | 91% | | 67% | 100% | |
| Maintenance and Repairs | 75% | 97% | | 50% | 100% | |
| Subprogram Pluvial and Sanitary Sewage | 0% | 0% | | 63% | 97% | |
| Infrastructure Works | 0% | 0% | | 63% | 97% | |
| Maintenance and Repairs | 0% | 0% | | 0% | 0% | |
| Public Works Management | 33% | 100% | | 25% | 100% | |
| Subprogram Construction and Maintenance Works and Urban Roads | 33% | 100% | | 25% | 100% | |
| Infrastructure Works | 0% | 0% | | 25% | 100% | |
| Maintenance and Repairs | 100% | 100% | | 25% | 100% | |
| Expropriation of Goods | 0% | 0% | | 0% | 0% | |

Source: PB of the municipal DAG of Gualaquiza.

### 3.2. Citizen Satisfaction and Perceptions

Surveys were used to gather information on the level of satisfaction and citizens' perceptions regarding the planning and execution of participatory budgets. The reliability level of the survey was validated with the Cronbach's alpha coefficient where a value of 0.90 was obtained, which qualifies the survey as a reliable instrument. Table 4 shows that opinions have been collected from citizens with different socioeconomic realities, therefore, with different perceptions.

**Table 4.** Characteristics of the surveyed citizens.

| Sample Characteristics | Percentage | | |
|---|---|---|---|
| Gender Female | | | 57% |
| Have a disability | | 4% | |
| Mestizo | 4% | | |
| Have no disability | | 53% | |
| Mestizo | 49% | | |
| Shuar | 4% | | |
| Gender Male | | | 43% |
| Have no disability | | 43% | |
| Mestizo | 43% | | |
| Total | 100% | 100% | 100% |

Source: Survey applied to communities.

With the data from the first part of the survey, it was determined that a total of 54% of citizens stated that the PB process is inclusive. This is aimed at citizens in general and the call is conducted through radio, but means to facilitate access to people with physical disabilities have not been established. Citizens who believe that the participatory budget is not inclusive (46%) expressed their disagreement by mentioning that notifications are provided with short notice, and that many citizens cannot attend due to their planned activities and working lifestyles in the countryside (e.g., they arrive home at night and do not know about the call).

Likewise, from the survey, it was determined that in regard to satisfaction levels, only 9% are satisfied with the execution of the works. A total of 91% of citizens are not happy because they do not know that the works are completed under this methodology. This lack of knowledge is due to the poor communication of the municipality with its citizens. In some cases, citizens do not know about planning because they did not attend meetings, or they simply forgot about it because no follow-up process exists, neither from the municipal authorities or technicians nor from the representatives of the communities. The low levels of participation in the assemblies are the cause of the low levels of satisfaction of the inhabitants. Per Holdo states that inviting people to participate in decision-making regarding the priorities of the municipality, especially when resources are scarce, helps build trust [61].

The second part of the survey, results are presented Table 5, provided insight into the perceptions of citizens regarding the PB implementation process. The "Community Planning and Participation" phase showed an average rating of "Disagree" because the citizens state that they have not been trained to participate in the process, nor do they believe that the municipality has established the necessary mechanisms to include people with physical disabilities in the assemblies.

For the "Execution" phase, citizen perception receives an average rating of "Indifferent", that is, they believe that both the process of reporting the start of works and meeting the deadlines for the completion of the works were executed in half or were included. In addition, for the "Follow-up" phase, the perception of citizens qualifies as "Disagree", especially due to the lack of support and reports from the municipality regarding the planned works.

According to the correlation analysis, shown in Table 6, a high level (0.81–0.99) is presented between the two variables: C1: "The municipal authorities have monitored the execution of the works" and C2: "The municipal authorities have come to report the completion and delivery of the work".

**Table 5.** Average responses to the survey.

| Questions | Average | Equivalent |
|---|---|---|
| Community Planning and Participation | 2 | Disagree |
| Did you know of the call to the PB assemblies? | 3 | Indifferent |
| Do you think that the calls to the PB assemblies were made by the appropriate means? | 3 | Indifferent |
| Were mechanisms implemented so that people with disabilities can attend the assemblies? | 2 | Disagree |
| Were you trained to participate in PB assemblies? | 2 | Disagree |
| Execution | 3 | Indifferent |
| Did the municipal authorities go to your community to report the start of the PB works? | 3 | Indifferent |
| Were the planned works completed within the established period? | 3 | Indifferent |
| Monitoring | 2 | Disagree |
| Did the municipal authorities permanently come to your community to report on the progress of the PB works? | 2 | Disagree |
| Did the municipal authorities go to your community to report the finalization of the PB works? | 2 | Disagree |
| Do you think that the implementation of PB has brought improvements in the quality of life in your community? | 3 | Indifferent |

**Table 6.** Correlation coefficient between survey questions.

| Questions | Community Planning and Participation | | | Execution | | Monitoring | | |
|---|---|---|---|---|---|---|---|---|
| | A2 | A3 | A4 | B1 | B2 | C1 | C2 | C3 |
| A　Community Planning and Participation | | | | | | | | |
| A1. Did you know of the call to the PB assemblies? | 0.74 | 0.49 | 0.16 | 0.24 | 0.20 | 0.58 | 0.60 | 0.22 |
| A2. Do you think that the calls to the PB assemblies were made by the appropriate means? | | 0.41 | 0.31 | 0.40 | 0.34 | 0.44 | 0.44 | 0.44 |
| A3. Were mechanisms implemented so that people with disabilities can attend the assemblies? | | | 0.60 | 0.48 | 0.59 | 0.67 | 0.61 | 0.43 |
| A4. Were you trained to participate in PB assemblies? | | | | 0.29 | 0.55 | 0.37 | 0.28 | 0.26 |
| B　Execution | | | | | | | | |
| B1. Did the municipal authorities go to your community to report the start of the PB works? | | | | | 0.78 | 0.62 | 0.51 | 0.76 |
| B2. Were the planned works completed within the established period? | | | | | | 0.60 | 0.46 | 0.61 |
| C　Monitoring | | | | | | | | |
| C1. Did the municipal authorities permanently come to your community to report on the progress of the PB works? | | | | | | | 0.92 | 0.59 |
| C2. Did the municipal authorities go to your community to report the finalization of the PB works? | | | | | | | | 0.61 |
| C3. Do you think that the implementation of PB has brought improvements in the quality of life in your community? | | | | | | | | |

Source: Survey applied to communities.

A medium–high correlation (0.60–0.80) was presented between the following variables: B1: "The municipal authorities went to their community to report the start of the work" has a coefficient of 0.78 with B2: "The works have been completed within the established term". Likewise, B1 has a correlation of 0.62 with C1: "The municipal authorities have monitored the execution of the works".

Furthermore, B1 has a correlation of 0.76 with C3: "The citizens consider that the works through PB have improved their quality of life". This demonstrates that when the authorities get involved in the process, the results are positive. A3: "The municipality has established mechanisms so that people with a disability can attend the assembly" has a ratio of 0.60 with A4: "Citizens have been trained to participate in assemblies". Likewise, A3 has a correlation of 0.76 with C:1 "The municipal authorities have monitored the execution of the works". Additionally, A3 has a correlation value of 0.61 with C2: "The municipal authorities have come to report the completion and delivery of the work". B2: "The works have been completed within the established term" has a coefficient relation of 0.60 with C1: "The municipal authorities have monitored the execution of the works", and a 0.61 correlation with C3: "The citizens consider that the works through PB have improved their quality of life". This demonstrates that when the authorities show interest in achieving greater inclusion, they are also there to accompany the entire process. A1: "The citizens knew about the summons to the assemblies" is related by a value of 0.74 with A2: "The citizens believe that the calls were made by the appropriate means".

The variables with the lowest correlation (0.01–0.20) are A1: "The citizens knew about the summons to the assemblies" with A4: "Citizens have been trained to participate in assemblies"; A1 with B2: "The works have been completed within the established term"; A1 with C3: "The citizens consider that the works through PB have improved their quality of life". In these questions, the incidence of one variable against the other could not be established.

## 4. Discussion

The successful implementation of PB must be based on compliance with the principles of citizen participation. In the municipal DAG of Gualaquiza, "Respect for Differences" is applied. "Responsibility and Coresponsibility" and "Popular Control, Information, and Transparency" are principles rarely considered by inhabitants who have little interest in participating in municipal management. Citizens tend to consider it as an unproductive activity with no visible results. Compliance with the principles of citizen participation expressed in Ecuadorian legislation is highly dependent on the political will of the elected representatives and authorities, who see PB as simply as an extra administrative process that has to be fulfilled.

At present, the idea of achieving governance without a government has gained strength, but through more participatory means [62]. Thus, as a result of the application of PB, improvements were expected, including increased government efficiency, organized expression of citizens, and a contribution to the improvement of the living conditions of the community [16]. However, achieving the objectives related to management and governance in rural areas is not always possible due to bureaucratic processes, poor connectivity, poor experiences, and the interest of both the residents and the municipal DAGs. The municipal DAG of Gualaquiza has a low capacity for spending; the percentage of compliance with the number of works planned through PB was 20% for 2017 and 43% for 2018. The amounts of PB spent were 29% (total spent/total budgeted) for 2017 and 28% for 2018. These results are lower than the levels of the entire municipality budget 69% in 2017 and 68% in 2018. These data demonstrate that, although the levels of municipal budget compliance are not ideal, the amounts granted through PB were applied deficiently due to the situations mentioned above. Therefore, it can be concluded that the PB mechanism has not been successful in the rural communities studied.

The scarce provision of participative education for the citizens is an additional feature of the low capacity of this municipality's spending. This situation is ultimately reflected in their perception that PB has not contributed to improving their quality of life, nor to the growing list of unsatisfied basic needs of the Ecuadorian population [63]. Additionally, citizens, consider PB as another bureaucratic activity and they do not have an interest in participating in the assemblies. In their limited participatory experience over a short time of application, these rural communities have not observed differences in their quality of life or in the management of the municipal DAG

The participatory political design of the Rafael Correa Citizen Revolution has improved democracy and citizen participation mechanisms. However, the level of citizen participation in 2018 remained quite low. According to the National Secretariat of Planning and Development, the total citizen political participation rate in socio–state interfaces is 3.07% [64]. The Citizen Revolution has generated changes in the country. The participation mechanisms create a broad and comprehensive system, but the limited application and lack of continuity of some mechanisms casts doubt on whether the process under development is sufficiently consolidated [65]. Monje [64] stated that such changes cannot be authentic structural changes; instead, they only relocate to an economic and political class, thereby strengthening only one political party [66]. This situation became evident following the change in government changed in 2018. Participatory activities in the communities have been reduced since then.

The findings found allow us to detect the aspects that must be changed to improve citizen participation and, therefore, for a better application of PB and an improvement in governance in the long term. PB, when applied in rural areas, presents a challenge in the search for community outreach tools. In order to improve the high level of dissatisfaction (91%) the municipal DAG could promote participatory education and strengthen communication with communities as a transparency mechanism. The municipality must commit to follow-up and submit continuous progress reports. In the PB implementation process, contact with the media must be constant and continuous—not only at the beginning of the process [67]. Calls must be made, and progress of the work must be published in newspapers, magazines, and advertised on the radio, and officials must have direct contact with citizens regularly and talk about the work completed by the local government to improve the planning and execution of programs, projects, and services. The application of participation mechanisms in Ecuador depends largely on the breadth and quality of the calls to assemblies made by state entities and the level of motivation of citizens who refuse to participate or are unaware of the process [65].

Communities require both the municipal DAG and community representatives to constantly consult and monitor the execution of their works. According to the correlation analysis, it has been determined that a community perceives better results when the authorities accompany the PB process in all its phases. This feedback will allow the participation cycle to be completed [15]. As information on the works executed is not publicized, citizens do not have the opportunity to ascertain the implementation of the works or to verify how the resources were used. Democratic governance is achieved when politicians allow access to participation and citizens have greater control over government acts [62]. For Barbera, Sicilia, and Steccolini, there are four conditions that must be met to successfully implement the participatory budget: responsiveness, representation, inclusion and interaction [68].

The findings prove that democracy and governance are not possible with political will alone but also require informed and trained citizens. Governance requires vigorous citizens that can become authorized voices in political decisions [9]; over time, these citizens will acquire power in their decisions [7]. Governance and participation are possible after the execution of a variety of disciplines, but they generally focus on institutional aspects and range from being methodologically pragmatic to very complex [69]. In order for PB to contribute to the improvement of governance, a broad participation of the citizens is required. Part of the success of PB lies in its educational functions with respect to the promotion of citizen participation in democratic issues. The experience of participating in the formulation of PB has had a positive relationship with an increase in improved skills and attitudes; Montambeault [70] has educated, trained, and even motivated constituents to demand more and increase the accountability of the government [71].

The results described above have been limited by the following: (1) The difficulty of carrying out a comparative analysis of previous years because the application of PB in the communities is recent, and the changes generated are not yet noticeable. The DAG has not generated an information system that allows monitoring the process continuously, which is why the best way to evaluate the process is qualitatively and through management indicators. (2) The difficulty in conducting participatory

research within the communities due to their remoteness and isolation. (3) The continuous change of municipal staff and elected representatives, which is a big obstacle when monitoring the process.

The next steps to follow in this line of research should focus on the determination of a methodology to evaluate the application of PB in rural communities with limited resources. The challenge is to define indicators and standards that adapt to the circumstances of these communities in way that allows for the measurement of the real impact of the application of both participatory mechanisms of governance and improvements in the quality of life of inhabitants, but with the inclusion of variables specific to the socioeconomic context of each community, especially rural communities.

It is observed that despite being the most-applied method, the single application of PB as a mechanism for improving governance is insufficient, especially in short and isolated periods of time. In order to guarantee a true citizen participation, PB needs to be complemented with other mechanisms such as popular consultations, social initiatives, referenda, revocation of the mandate, political deliberation, local assemblies, public hearings, popular councils, sectoral citizen councils, advisory councils, local planning councils, national councils for equality, previous consultation, empty chair, citizen observatories, and citizen oversight [40]. These mechanisms have been complemented by the possibilities of digital technology in view of the growing demand for individual involvement in participatory innovations aimed at effective governance [72].

**Author Contributions:** Conceptualization, I.B. and P.V.; methodology, I.B. and F.C.; software, I.B. and F.C.; validation, P.V. and J.L.Y.; formal analysis, I.B.; investigation, I.B.; resources, I.B.; data curation, I.B.; writing—original draft preparation, I.B.; writing—review and editing, P.V.; visualization, I.B.; supervision, J.L.Y.; project administration, I.B.; funding acquisition, I.B. All authors have read and agreed to the published version of the manuscript.

**Funding:** This research received no external funding.

**Acknowledgments:** We thank the Universidad Politécnica Salesiana of Ecuador for the funds provided for the information survey and the Municipal DAG of Gualaquiza for the access granted to the Participatory Budgeting documentation.

**Conflicts of Interest:** The authors declare no conflict of interest. The funders had no role in the design of the study; in the collection, analyses, or interpretation of data; in the writing of the manuscript, or in the decision to publish the results.

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
