# Peer review of "The Participatory Budgeting and Its contribution to Local Management and Governance: Review of Experience of Rural Communities from the Ecuadorian Amazon Rainforest"

_sustainability, doi:10.3390/su12114659_

Round 1
Reviewer 1 Report
Please, find suggestions for revisions directly in the file. I think that after addressing these minor things the paper is ready for publication. Good luck!
Kind regards

Author Response
Dear Reviewer, we welcome your suggestions. We send the comments in the PDF document.

Reviewer 2 Report
Although some of the weaknesses indicated in the first review have been improved, the result is still unsatisfactory due to the insufficient information that would be relevant tto obtain an evaluation of greater scope and depth of PB Program. Both the characteristics of the case study and the scarce temporal depth that the research embraces prevent obtaining a more adjusted perspective of the incidence of PB in the governance and local management of the municipalities studied. But most important of all is the lack of information on the behaviors, attitudes and practices of the actors involved, both the local population, and political decision-makers and administrative personnel, information that is impossible to obtain through the data collection instruments used. This makes the results offer a too flat perspective, with conclusions of a very general nature and very little operative in view of the possible improvement of the PB processes, the greater involvement of the local population and the increase in its incidence in the necessary changes for the betterment of community life and governance.
With respect to the surveys, which are reported to have been carried out again, some more information is provided, but not on the selection criteria of the sample, but on the characteristics of the people who responded, revealing what, I understand , are important imbalances, both in terms of gender distribution, and, especially, by the homogeneity of those carried out for men (non-disability and mestizo) compared to that answered by women, with a greater diversity of variables (disability / no-disability, mestizo / shuar).
Likewise, the reasons for the telephone or face-to-face nature of conducting the surveys are not clear, nor are the number and distribution of both. All data that I consider relevant to assess the reliability of the results.
Nor is clear the nature and type of the interviews conducted with mayors, community leaders, and technicians. Are they part of the surveys or were they carried out independently? And in this second case, of what type were the interviews and with what script or questionnaire?
Author Response

(The authors gave the same response as above.)

Reviewer 3 Report
The paper has been greatly improved, it is better articulated and much better documented. Well done!
Some remarks and comments in order to help the authors improve it further:
L. 48-52 It is a sentence rather difficult to understand (more than 70 words in one sentence). Please try a simpler construction.
L. 68 – 69 In addition to stability and efficiency, the favoring of government by the citizens adds legitimacy to the processes
What do the authors mean by this phrase? A government not favoured by citizens lacks legitimisation anyway.
L. 94 -95 As such, the bureaucratic centralism of bankruptcy is managed through decisions shared between the state and society.
What exactly do the authors mean by that phrase?
L. 144- 159 I fail to understand the function of this part in the manuscript
L. 288,289 surveys
Do the authors mean responses, completed questionnaires, interviews? The survey was only one in two parts.
L. 363-367 How did the authors derive this statement?
L. 382-387 How did the authors derive these statements?
Furthermore: The low levels of participation in the assemblies are the cause of the low levels of satisfaction of the inhabitants.
One could argue for the reverse. Low levels of participation could be attributed to low levels of satisfaction by the work done.
L. 402 Table 8
Which correlation coefficient did the authors use, bearing in mind that use of a Likert scale does not produce a continuous variable?
Author Response

(The authors gave the same response as above.)

Round 2
Reviewer 2 Report
Dear authors, I appreciate your effort to improve the text, but as you recognize, there are significant difficulties to overcome most of the shortcomings and limitations that, in my opinion, affect the solidity of the article, but the difficulties cannot be an argument to justify the weakness of the data on which the analysis is intended to be carried out, with which, if they are insurmountable (short time span, dispersion of communities, difficulty in accessing certain population groups or type of informants, ...), perhaps the The most scientifically consistent option is to give up carrying it out, at least until better conditions can be found.